# Aesthetic Enactment: Engagement with Art Evoking Traumatic Loss

**Lynn Froggett [1],* and Jill Bennett [2]**

[1] Department of Social Work Care and Community, University of Central Lancashire, Preston PR1 2HE, UK
[2] Big Anxiety Research Centre, University of New South Wales, Sydney, NSW 2021, Australia; j.bennett@unsw.edu.au
* Correspondence: lfroggett@uclan.ac.uk

**Abstract:** This article analyses audience responses to two creative works inspired by traumatic loss—the first, a performance presentation, recounting events from the author's adolescence; the second, a short film about a suicide in the filmmaker's family. Both were shown in 2017 as part of a mental health arts festival, attracting audiences with affinity for the lived experiences portrayed. Given the potential for such works to give rise to negative feelings and/or to retrigger trauma, the objective of this research was to understand firstly whether audiences could process the trauma conveyed in a contained and facilitative setting and, secondly, how the specific aesthetic modality of each work supported this processing. The psychosocial methodology adopted consisted of a group-based, image-led associative method—the visual matrix—which invites participants to express their sensory-affective and felt responses to a creativework. In the case of both works, the visual matrix gave rise to a distinctive form of *aesthetic enactment*, expressed through rhythm and image association. This imagistic and 'rhythmic' mode of engagement appeared to be key to the re-symbolisation of trauma for the audiences. The implication of this study is that the re-visiting of potentially distressing experiences in an aesthetically mediated, containing setting is potentially reparative in its effect.

**Keywords:** aesthetic; performance; suicide; trauma; OCD; enactment; psychoanalysis; psychosocial; rhythm; visual matrix; transitional space

## 1. Introduction

Narrative film and performance arts have long played an important role in examining human experiences of trauma, loss, and psychological distress. Here, we consider the public screening and staging of artworks that present troubling or traumatic material in a context where audiences are likely to have lived experiences that relate to those depicted. We take our examples from a study of audience engagement that shows how the primary material of a film and performance lecture, which in themselves contained no uplift and little ground for optimism, is enacted and creatively transformed by the viewers in particular conditions. Specifically, we examine how a form of "third space" or shared "transitional space" (Winnicott [1971] 1991; Benjamin 2018) is produced in a visual matrix setting (Froggett et al. 2015) and how this enables participants to use and reflect on traumatic experiences and potentially transform them.

The works that we consider were shown in the context of "The Big Anxiety", a public mental health and arts festival in Sydney in 2017. The Big Anxiety addresses a wide range of mental health experiences, with a primary focus on trauma and on community and lived experience perspectives (Bennett 2022). We were interested in whether and how the presentation of these artworks could facilitate a reparative processing of the emotions attached to mental distress, trauma, and loss, at least in the short-term. Audience members were self-selected, for the most part unknown to one another, and those who took up the offer of a visual matrix were accepting an opportunity to re-visit and make sense of an aesthetic experience in the company of others. We studied their responses in the

immediate aftermath of the viewing experience. As such, we focus on the reparative potential of aesthetic engagement (Segal 1991) rather than on tracking longer-term or sustained reparative effects.

By its very nature, the festival attracts a public with an active interest in mental health and identification with the experiences presented. The Big Anxiety is open to all, without any filtering of attendance or universal collection of data on the mental health profiles of people who choose to come. It is therefore of some importance to carefully curate the conditions under which work is presented (Bennett et al. 2019a) and to understand the potentially reparative or harmful effects of showing troubling artworks in public contexts. We had previously used the visual matrix in the qualitative evaluation of artworks (Froggett et al. 2015; Bennett et al. 2019b) and were aware of the high level of containment (Bion 1970) it offers. By using the visual matrix to work with the audiences, we hoped to see the audience at work (Froggett et al. 2019), unconsciously enacting, playing, and re-symbolising their experiences. Our specific aim in this instance was to see whether the scope for sensory affective expression that it enables in a supportive group context might enable the audience to accommodate or work creatively with their experience and, if so, how and to what end.

In this article, we analyse audience engagement with two contrasting works: one, an experimental video and live performance relating to a story of trauma and self-professed "OCD" (dis/ordered), staged at the Museum of Contemporary Art, Sydney; the other, a short film about a rural Australian family in the immediate aftermath of a suicide (The Invisible Edge), screened for the public at a mental health research institute. The Big Anxiety festival operates within a distinctively psychosocial trauma-informed support protocol, placing emphasis on safety and containment with opportunities to debrief, talk informally, and/or to connect with counsellors or designated support staff after the engagement. This prioritises cultural rather than clinical framing and trauma-sensitive facilitation rather than medicalisation (Bennett et al. 2023). In the case of the film screening in the institute, mental health professionals were available. The lead facilitator was a trained mental health social worker.

In each case, having viewed the work, the audience was invited to take part in a visual matrix as a means of formative facilitation designed to offer a safe space for sharing responses. The process promotes association rather than analysis or explanation, and it facilitates exploration of feelings and impressions rather than judgements. The aim here was to capture affective responses as well as articulated thoughts, so as to better understand the reparative possibilities of aesthetic engagement. This is an important consideration, as distress and trauma are not always fully available for discursive representation and description. This method enables us to track what happens when troubling sensations are experienced, and to detect the potential for harm where people might find the material confronting or "triggering"; also, to assess the extent to which distress or troubling memories are "metabolised" and symbolised (Bion 1970). Jessica Benjamin (2018) points to the phenomenon of enactment of traumatic or troubling experience as a form of communication leading to play. This offers a theoretical framework within which to understand its reparative potential. We return to this matter in the Discussion.

## 2. Method

### 2.1. The Visual Matrix

A visual matrix typically accommodates between 10 and 30 members and takes place immediately after a viewing of stimulus material (in this case, the live performance lecture and the film screening). The matrix itself runs for 45 min to an hour, followed by a short break and then a discussion session of up to an hour, in which participants collaborate on preliminary analysis, setting the frame for the interpretive work of the researchers/facilitators that will follow. During the matrix, participants are invited to offer images, thoughts, and feelings aroused by the stimulus or by one another's associations, as and when they want, and without turn-taking. The facilitation elicits image-led, af-

fectively laden associations rather than discussion or evaluation. Participants are seated in a snowflake configuration, conducive to speaking into a shared space, rather than addressing one another (Froggett et al. 2015). This set-up also helps to avoid the alliances, confrontations, and power imbalances typical of group dynamics. The subsequent group discussion (for which chairs are rearranged in a circle) allows the participants to reflect on the matrix process and their experience within it, mapping its key clusters of imagery, thematic content, and shifting affects.

As an image-led, group-based psychosocial method, which elicits responses to a visual/aesthetic stimulus, the visual matrix method takes inspiration from social dreaming (Lawrence 2005), and also from Bion's understanding of reverie (Bion 1970), which is the condition for the associative process[1]. Participants' contributions are often dream-like, poetic, and infused by visual/sensory metaphor. The matrix itself offers a transitional space (Winnicott [1971] 1991), allowing participants an opportunity to reconcile subjective experience with the inter-subjective reality that is acquiring symbolic form in the shared visual/sensory collage of associations that the participants produce.

### 2.2. Data Analysis

This is a qualitative study, in which self-reflexive researcher observation and interpretation is used throughout, building on an interpretative frame set by the participants. As such, researcher responses to the data are actively deployed and safeguards against over-interpretation are essential (see Froggett et al. 2015) for a detailed discussion of interpretive protocols and their depth hermeneutic theoretical underpinnings in the work of cultural analyst Alfred Lorenzer (1986; Salling Olesen 2012). In summary, the process follows these steps:

1.  The interpretive frame is established in a post-matrix discussion by the participants themselves as they reflect upon what emerged in the matrix, focusing on the dominant themes and affective intensities.
2.  Research team panel analysis is used in which analysts first discuss their own responses to the data, listen to the recordings, and annotate the transcript with their observations.
3.  The panel members discuss and contest one another's interpretations until saturation is reached.
4.  Provisional hypotheses are generated as the research analytic teamwork through the transcript and recording. To "survive", a hypothesis needs to find iterative support as the team works through the dataset.
5.  The depth hermeneutic protocols deployed consider the data sequentially in three modes: substantive (what was said or presented), performative (how it was said or presented), and explanatory (why it was said or presented thus). There has to be some triangulation or coherence between these modalities to arrive at an interpretation that is considered to be well supported by the data.
6.  The research team remains "close" to the data, working not only from a transcript but repeatedly returning to the original audio-recording and discussing and comparing their own emotional responses and disposition to interpret in particular ways. The aim is to maintain self-reflexivity while ascertaining how utterances were enacted (performatively), and also the shifting affects of the matrix as a whole. This distinctive feature of the method yields a richer perspective than thematic analysis alone and allows the research team to apprehend unarticulated and unconscious aspects of the process.

### 2.3. Aesthetic Engagement

We suggest that engagement with creative works facilitates a particular kind of "third space" or "transitional space" wherein participants can operate in an aesthetic modality. Through our examples, we explore how each visual matrix tends to develop its own distinctive aesthetic, which, in important respects, echoes and elaborates the aesthetic of

the artwork to which it responds. This is a form of unconscious enactment on the part of a working matrix.

Our focus is on the nature and quality of aesthetic engagement (Bennett and Froggett 2020). Whereas most forms of audience response inquiry ask what an audience thought, translated and reported in non-aesthetic terms, the visual matrix allows us to investigate how an art production enables an audience to take up and make use of an aesthetic idiom. Each of our two matrices distinctively reflected the styles of their stimulus: one, a "performance lecture", accompanied by a fast-paced video mash-up, drawing liberally on popular culture, TV, and YouTube sources to construct a narrative that questions the social determinants of mental health; the other, a slow ambiguous film, dramatising a young man's return home after his brother's suicide.

We found that, in both cases, the audience in the matrix responded to what they had witnessed imagistically and metaphorically through their own figurative associations. We observed them spontaneously enacting the production's idiom so that the matrix was in important respects *isomorphic* with the live performance or film they had viewed. Their response, in other words, was not merely *to* an aesthetic but was effectively enacted *in* an embodied aesthetic mode. We describe how these enactments involved a reparative re-symbolisation of the troubling material of the artworks rather than "re-triggering" trauma or distress. The matrix that creates the conditions for enactment "or enactive play" (Reis 2019; Benjamin 2018) thus allows us to examine how pre-reflective attunement to aesthetic qualities of the artwork itself—to its rhythm, flow, soundscape, and palette—supports participants to work creatively with the experiences portrayed without being overwhelmed.

## 3. *Dis/Ordered*—Matrix 1

### 3.1. The Stimulus

Clive Parkinson's solo performance lecture, *dis/ordered*, recounts autobiographical experience of personal trauma and a life lived in its wake. A scripted narrative is performed live by Parkinson, synched to a video backdrop, the style of which is a fast-paced montage of contemporary and historical media clips. These combine to tell a story that is intensely personal in its focus on events in the decaying seaside townscape of Morecambe on England's North-West seaboard. The second part is more overtly political in its broader social critique of neoliberal healthcare systems and the shortcomings of their treatment regimes and models of care.

Embedded within the grand sweep of a narrative spanning a century of Morecambe's prominence as a seaside resort is a single momentous event from Parkinson's youth:

> *On July 10th 1981. . .fuelled by cheap cider, glue and infinite possibilities, three mutual friends set out to cross the 15-mile bay.*

These friends, Parkinson tells us, were never to return:

> *The first body was found face down in the Harbour the following week, the second, and my closest friend of the three, was found 18 days after they'd set sail, in a picnic spot I still frequent. The third, to my knowledge, was never found.*

The boys' stolen boat "emblazoned with the yellow number four" becomes emblematic of the traumatic event, and a portent of the theme of numbers, counting, weighing, and measuring. The fetishisation of numbers is echoed in both images of children's games that assign significance ("one, two, buckle my shoe; three, four knock on my door") and a more sinister imagery conveying the relationship between metricisation, population control, and economics, referencing Francis Galton's anthropometrics.

The tragedy that haunts the entire work is little elaborated within Parkinson's narrative. Instead, the performance enacts and reflects on the way that trauma is unspoken and often mutely embodied. In Parkinson's telling, it gives rise to the behaviours that characterise obsession and the compulsion to repeat, here understood not as a mental "disorder" but as a traumatised adolescent's way of imposing order on chaos or a "disordered" environment.

Thus, *dis/ordered* is described in publicity material as "questioning diagnosis and exploring the ways in which people make sense of seemingly intrusive rituals and behaviour".

### 3.2. Repetition/Enactment

It is important to consider how an enactment or repetition of trauma within an aesthetic framework offers a way to work with that experience. In this context, it does so through the artist's self-conscious play, which in the visual matrix evokes in the audience a largely unconscious response inducing them to draw on their own experiences to re-present and re-enact the compulsion that is being staged.

Whereas Freud considered that the repetition compulsion, which is impelled by an unconscious memory of traumatic experience, wholly negates or overrides the pleasure principle (Bollas 1987), later authors have stressed that repetition yields the psychological satisfaction of mastery (Fenichel 1946). However, Parkinson's performance piece demonstrates another possibility where "OCD" behaviours emerge in the form of children's games (never walking on cracks, obsessive counting). These are compulsive, self-soothing actions that in their playfulness suggest a working through via enactment and presentational re-symbolisation (through the verse and the game) that differs from a purely neurotic repetition (Reis 2019, p. 105). Reis proposes that when repetition of trauma is linked to a creative act of invention in this way, the task of psychotherapeutic work is to engage with this enactive play and focus not on uncovering what is repeated or re-lived, but on creating the conditions for new experience (Reis 2019). This distinction, we suggest, illuminates the (quasi-therapeutic) register of art, which in this case does not strive to *represent* trauma (the traumatic event, in fact, is referenced only briefly); rather, Parkinson reflects on—and in part enacts—his childhood compulsions, placing these within the frame of a performance/video work that gives rise to new kinds of experience not only for the artist but potentially also for the audience. This recomposition, using elements of past experience, effectively embodies the generative process, theorised by Bollas:

The work of trauma will be to collect disturbing experience into the network of a traumatic experience (now a memory and an unconscious idea), while the play work of genera will be to collect units of received experience that interanimate towards a new way of perceiving things. (Bollas 1992, p. 78).

### 3.3. Audience Engagement: Enactment and Metabolisation

Here, we are specifically interested in how audiences partake of this process, and how a felt connection to the content of the work and to the experiences described and evoked by it—here facilitated by a participatory visual matrix—enables the metabolisation and transformation of such experience (Bion 1970).

Invited at the inception of the matrix to offer an image or association promoted by the work, the participants start by referencing the film's imagery of drug taking, which includes footage of an acid-tripping party crowd from the 1960s, as well as more ominous imagery of pharmacological intervention and sedation. The matrix participants present these associations in the form of feelings of disgust and overwhelm:

- *This image is really strong, I found it very strong. . .makes me feel like it's like some kind of drug, a party, you know, parties for people taking drugs and they are like sick and dying. And just so disgusting and I couldn't stand it, I just close my eyes. That's really strong.*
- *I got an image of a comatose person who's been heavily sedated with psychiatric medication.*
- *I had feelings of frustration.*
- *I was overwhelmed by the word 'capital' and the repetition of capital letters [in super titles appearing on screen].*

The lines that follow these are similarly marked by the speakers' tendency to place themselves within the process (*I remain with, I visualised, I keep coming back*), and by an immediate immersion into the flow of association. These "strong" and perturbed impressions

are followed immediately by personal recollections, clearly identifying with the child and the compulsion to create order through counting:

- *I had images of my own less than joyful school experience.*
- *I have a memory of playing with lego when I was child and not really playing as much as making sure it was in the right order and then making sure it didn't go out of the right order.*
- *I saw repeated images of the sea, the waves that keep repeating well, one wave after another.*
- *Yes, I remain with the waves and I started counting to four as those waves came in.*

For the matrix participants, there is a progression *within* the counting compulsion that offers a partial release—a noticeable shift in tone here to a more relaxed, overtly playful engagement with the childlike pleasure in favourite numbers:

- *A whole lot of numbers: one, two, three, four. What struck me now is the numbers and the ideas that are words. I don't quite get these words whether they are just jumbled right now".*
- *I like the number three better than the number four. Number three is my favourite. moved on to eight.*
- *I don't like eight at all.*
- *I liked how two plus two equals five. . ..*
- *Greater than the sum of its parts.*

Typical of this matrix is a sense of oppression and disturbance which is fully acknowledged before pivoting to a more pleasurable affect. There is an oscillation between imagery associated with submission, which "like resistance is anti-growth, a masochistic giving oneself up" (Ghent 1990) and surrender "a force towards growth". The surrender enacted through collaborative play gives rise to what Benjamin calls the "Third", a space "beyond doer and done to" where participants can escape such designations, "giving over to a co-created structure that transcends and absorbs the individuals so that they attain a freedom from self-consciousness, effort or strain" (Benjamin 2018, p. 170). Likewise, in the matrix, images associated with being "done to" or with submission are described and identified in terms of their perturbing effects. However, these are contained by the matrix itself enabling the shift to a form of play, itself marking the readiness of the group to work together in generating alternative images. In departing from the presented imagery, the matrix begins to elaborate new material of its own that is not featured in the film but resonates with its ambivalent return to the 1970s and 1980s. An image of the electric bar heater takes shape, bringing pleasure in shared nostalgia, even though it turns on a sense of danger:

- *I remember my childhood and I left the bar heater on a cold winter's morning about breakfast time.*
- *It makes me wonder if anyone ever burnt themselves. Bar heaters are really hazardous.*
- *That reminds me of when the dishwasher caught fire.*
- *And my neighbours put a mattress up against their house and the light was on and then the mattress caught on fire.*
- *Someone getting pushed and falling onto the heater.*
- *I had a bar heater until the dog's tail got caught on it and burst into flames.*

The images and sensations of *dis/ordered* are here extended by the matrix—still faithful to the retro aesthetic of the performance—and re-combined with other ideas evoking the homeliness and threat of childhood environments. This shared creative production strikingly reverberates with the rhythm and deadpan delivery of the performance. Parkinson adopts an even intonation that somewhat flattens emotionality, moving with ease from the mundane to the disturbing with a self-aware, dark, or ironic humour. Horror and trauma surface within the banal everyday of *dis/ordered* without derailing the narrative flow. Similarly, the matrix works with a humour that verges on the grotesque, signifying disruption and alterity (the tail on fire should not be amusing, but somehow is), thereby proliferating associations in the modality of *dis/ordered*. There are, in this regard, two

intertwined dimensions to the performative isomorphism developing in the matrix: the one reflecting the fast, connective flow of the video montage; the other, the organising commentary of the live narrator whose story is being told.

Resonating with Parkinson's anchoring voice as the narrator, the matrix manifests a growing confidence in articulating the felt experience of being caught up in the sweep of history with its mechanisms of control and measurement and powerful forces. It is also energised by its own continuing creativity in the proliferation of new material. In the next phase, for two or three minutes, participants range over perceived "OCD" traits (folding clothes before putting them into the wash); three drowning punks, and a bloated body in water (starkly referencing the central tragedy of the story); inundations that wipe out habitats and livelihoods (Caribbean hurricanes and Atlantic weather fronts); global warming and the spectre of Donald Trump; selfie culture and Kim Kardashian—and images of asylums past and present, oscillating between associations of care and those of control and domination:

> *I've got conflicted views on asylums. Sometimes I see really amazing things, so not just the horror.*

### 3.4. Rhythmic Entrainment

Disparate images are seamlessly woven together, without pause, in an associative flow that never falters, echoing the relatively unrestrained and irreverent video mash-up of *dis/ordered*. The production itself (the video, overlaid with personal narrative) in this sense provides the aesthetic that the matrix re-enacts, not just in visual style but also in rhythm. In groups, as Teresa Brennan argues in *The Transmission of Affect*, rhythm is a means of affective entrainment ("the driving effect one nervous system has on another", Brennan 2004, p. 70). Responsiveness to rhythm is thus a binding force; it has "a unifying, regulating role in affective exchanges" with the capacity to transmit complex states. Benjamin, likewise, describes the experience of the Third as a "rhythmic" one, "harmonizing of different voices, infusing the symbolic elements of the shared reverie with vitality" (Benjamin 2018, p. 164).

In the matrix, we see shared reverie in and through a rhythmic, aesthetic mode, effectively mirroring the aesthetic stimulus. This kind of enactment is pre-reflective—a surrender to flow—but it also links to Benjamin's "moral third", witnessing and "acknowledging violation", and the "repair of breakdowns in lawfulness" (Benjamin 2018, p. 225). It examines *experientially* the tension between opposites: doer and done to; disgust and fascination; good and bad; freedom and domination—antinomies that surface and are held in balance throughout the matrix.

In the second part of the performance, Parkinson shifts into polemic. Rhythmically, this is a punchy, faster-paced delivery with undertones of anger or aggression, as he attacks the power and manipulation of media, of mental health diagnosis, and of the neoliberal economics that govern health provision. The politics is overt but a felt experience of being "done to" is pre-eminent:

> – *I felt sadness for the little boy who had difficulties with his reading and the teachers labelled him as stupid.*
> – *Yeah, as stupid, 'an idiot' I think was said. And when all he needed was some assistance with his reading and his whole life may have been very different as a result of that intervention.*
> – *I find it's [hard] to know that one's life been dominated or been determined by the media. . .So they're sort of controlling our imagination, our education and direct it exactly the way they want it. . .*
> – *the Bible and the DSM. . .both are red brown books, both are very thick.*
> – *Both are written by white men.*
> – *And both are controlling. . .*

Here, feeling for the "done to" figure of the child prompts expressions of opposition to powerful forces. Religion and science are brought into the same frame through the conflated authority of the DSM and the Bible, august leather-bound volumes both in their

own way objects of projected fantasy. The playfully imagined disintegration of these tomes focuses on "red rot" afflicting old, archived leather, reflecting the moribund authority with which they are invested, ending in "dust and earth". This is a cue for a meditation on death, decay, and decomposition, conjuring the tension between beauty and toxicity:

–    *. . .how beautiful Moreton Bay actually is. It's beautiful but it's toxic.*
–    *Reminds me of the Hunger Games. Everyone's beautifully dressed, there's no emotions.*

By now, the matrix is at ease with its own creative process and segues into a discussion of death beds and the personally experienced deaths of family members. This affectively laden imagery is ultimately staunched by a new image that does not derive from *dis/ordered*—that of "Babushkas" or "Russian nest eggs". This literal image of containment, pregnancy, and birth in the form of a child's toy again marks the group's capacity to "hold" what is confronting and disturbing in the performance, and then to co-create "a cushion against impingement" (Wright 2009, p. 33). Impingement in this context relates to the negative, subjectively diminishing experiences of which Parkinson provides copious illustration—from the initial images of drugged/comatose bodies, and a traumatised child, to images of death and destruction, the ascriptions of "stupidity", or of OCD and the politically charged bibles and diagnostic manuals seen as instruments of population control.

Following Benjamin, we suggest that the subjecting forms of social dependency are dislodged through engagement with the micro-affective interactions and "impingements" that effect recognition—which is to say, through an *aesthetic modality*. This engagement occurs in the matrix through its take-up and elaboration of Parkinson's own subversive attitude, rather than his overt message. It is for this reason perhaps that the attention of the matrix remains overwhelmingly on the first part of the performance rather than the second, where the "message" and its socio-historical context is more directly unfolded. The matrix embodies an attitude, making use of imagery within and beyond the performance to generate an aesthetic of felt experience. It accepts an invitation to subvert vertical authority (religion, school, family, and the mental health system) through the counterpoint of child's play (number games and counting) and co-created imaginative work. By this process, in Benjamin's words, "identifications that function as submission in one register can be reconfigured in the intersubjective register of thirdness".

## 4. *The Invisible Edge*—Matrix 2

### 4.1. The Stimulus

The short film, *The Invisible Edge*, is a narrative fiction, based on real events in filmmaker Ian Thompson's extended family in rural New South Wales. It centres on the story of two brothers whose lives have taken different directions. David, a pianist, returns to the family farm, tormented by the guilt that he feels about his brother Jack, a farmer, who has died by suicide. The narrative structure, which makes extensive use of flashbacks, intentionally generates ambiguity around these two figures, suggesting initially that David may be the one about to kill himself, with Jack occupying the position of larger-than-life and robustly stable brother. The twist by which David's medicated depression turns out to be grief for his brother rather than an indicator of his own suicidal intentions disrupts any stereotypical reading of male strength and distress. It gradually transpires to the audience that Jack is a ghostly figure within an internal dialogue that hints at entrapment in destructively entangled family relations and way of life—from which in the end there is no escape. *The Invisible Edge* is a slow-moving, reverential film with a difficult topic. The viewer is slowed down in its convolutions. There is little uplift, no happy ending, and the grounds for hope are uncertain. The audience is obliged to dwell in the emotional ambiguities of the storyline but is relatively unsupported in navigating them.

The confusion in the narrative as to which brother kills himself is not clarified to the first-time viewer until the very end of the film, but the audience that took part in the visual matrix had just seen the whole film and had had the outcome revealed to them. Here, we discuss how, in the conditions of the matrix, they re-immersed themselves in its sensory

and affective universe—as if by re-visiting the "feel" of what they had witnessed they could share and process their experience of it.

*4.2. Audience Enactment*

The opening words of the matrix are one-word descriptions with long silences in between that from the outset reproduce the film's tempo. They speak of "drowning", "heaviness", "isolation", "desolation"—single word interjections that convey a sense of the speaker losing vitality under an oppressive weight.

Very unusually, the first dozen or so contributions take a full six minutes reflecting the halting pace of the narrative development, and also the fact that aspects of the story remain baffling even at the end. The silences between utterances feel impenetrable, yet they seem to be performing important work. Sense comes in fragments. It is far from clear where the difficulty in the family is lodged—with which brother? with the mother? in the relations between them all? in the isolation and harshness of rural life?

Someone in the matrix refers to "the hollow sound of walking on floorboards" which amplifies the sense of threat in the film without ever clearly identifying the source. Elements of the physical setting are highlighted: smell, touch, vistas of lonely, baked countryside, the feel of the rain, heat, scratchy grass. In this way, the matrix resonates with the film's sensory repertoire—its tempo, palette, and landscape, thereby taking up its mood and "language" as an artwork.

Many of the images that follow in the matrix are of being shackled and chained, tormented, and constrained by duty and obligation. There are also a few references to love, strength, willpower, growth, and regeneration, but hope appears "blind". A strong tendency to polarised emotion dominates the first half: reaching out and holding back, expansiveness and constraint, heat and cold, letting go and holding on, joy and pain, strength and weakness, safety and risk, push and pull, entrapment and freedom, childhood and adulthood, life and death, past and present.

However, the urge in the matrix is towards reparation and this is first revealed in the way that the tempo counterbalances the binaries by spontaneously organising utterances into "stanzas" of between three and five related ideas separated by long reflective pauses. These pauses allow each cluster to be digested before turning to the next associative sequence, which often presents a contrast:

– *Angriness and yelling.*
– *The burden of family expectations.*
– *Collision.*
– *Brothers.*
– *Torment.*

*[long pause]*

– *Vulnerability and the kindness of the stranger, of the old school friend.*
– *Reaching out.*
– *The beauty of the landscape.*

The film itself avoids the spurious gratification of polarised emotion and of easy resolution, preferring to confound explanatory links in the narrative. Instead, it uses the screenplay and soundtrack to pitch the audience into a world freighted with foreboding and loss. What is beautiful in the landscape is also desolate and isolating. The pacing remains careful and deliberate, designed to give audiences time for sense-making. The matrix takes up this invitation, allowing lengthy intervals between contributions and making use of the in-between spaces to acknowledge disturbance and "try out" fragile moments of consolation.

In the "stanzas" quoted above, the long pause enables a conversion from the "burden of family. . ." *to "the kindness of the stranger"; from "collision" to "reaching out".* Manic flight into unfounded hope is avoided. The momentary calm of the landscape gives way once again to danger—hinting at the conflagration of the bushfire and human terror:

- *Smouldering.*
- *The smell of eucalyptus.*
- *I think I smelled male sweat—smelling of fear and panic.*

Throughout the matrix, the imagery that gives rise to the film's sense of devastation is pored over and considered in long passages of silence. Ideas take time to "ripen" into speech. The silences continue to reflect (but not imitate) the still spaces in the film. They are unusually long—up to a minute or more—but these are moments of pause rather than blockage and they continue until the very end of the matrix. As participants remark in the post-matrix discussion:

- *And I mean, silence is part of the problem, right? Like, there is a lack of talking about, like, you know, where is the—where is the emotional outlet? Where can I—where is there room for me, like, and that place is silent and there is no noise or busyness or interconnection, so. . ..*
- *Yes, that's right. Yes. So, it's almost as if there was some sort of reparation going on in those silences then because these were not silences with no connection.*
- *Yeah.*
- *People always found their thoughts within the silences and then proceeded with them.*

At a perceptual level, we can see the silences (in the film itself and the matrix) as "stop" points. In the matrix, this is where members note what matters in the presence of others. Applebaum's (1995) concept of "the stop" is in reference to occasions or events where we are stopped in our trajectory and come to the awareness of other yet-unrealised possibilities: this stop moment is an embodied shift—a moment of risk, a moment of opportunity. Applebaum (1995) writes: "between closing and beginning lives a gap, a caesura, a discontinuity. The between-ness is a hinge that belongs to neither one nor the other. It is neither poised nor unpoised yet moves both ways".

The silences are needed for the group to find and process their thoughts, but ideas once uttered result in another flight of association. Silence in the film is part of the problem—the silence of the place and the collusive silence surrounding the suicide. Silences in the transitional space of the matrix, by contrast, increasingly become transformative interludes where new imagery and ideas emerge and take shape. The work of the silences can be understood in the context of Winnicott's ([1971] 1991) notion of transitional space—a pre-symbolic space "in-between" feeling and form—as an area of experiencing where transitional phenomena occur. These are precursors to symbolisation that takes shape in the audience's creative production of image associations. Strikingly, the long silences continue to the very end of the matrix as spaces that afford "creative finding" of needed forms that pre-figure the possibility of a future:

- *And thinking about burning the whole farm down so that they can all be free.*
- *Like those seed pods in the bush that need fire to open.*
- *Growth and regeneration.*

The transitional or third area of experiencing (Winnicott [1971] 1991) is where new experiences can emerge and find resonating cultural forms for their expression. If we speak of the participants using the matrix as a transitional space to "process" as yet unarticulated experience of an artwork or performance, it is the finding of sensory forms for this experience that is at issue.

When functioning optimally, there is a strong sense of the matrix resonating to presented imagery and temporal flow, whether from the stimulus, or to the biographically rooted imagery and thoughts of participants that are triggered in the matrix and reverberate with one another. This is not simply a mirroring process, although there are mimetic moments as when imagery from the film is re-presented unmodified. Aspects of *The Invisible Edge's* landscape and "mood" provide examples.

*The contrast of the beautiful countryside and the tragedy that happened with it all.*

However, purely descriptive statements like this are relatively few and far between. Instead, the matrix transforms or "re-casts" the experience of the film into new images and

thoughts which partake of what the original stimulus presents, and also of what arises in the minds of the participants. For this to happen in *The Invisible Edge*, there has to be attunement in the matrix (Wright 2009) to the affective and aesthetic quality of the film and a responsiveness towards its symbolic repertoire. It is from this attuned re-casting that the isomorphism of the matrix with the film arises. In the following example, the soundtrack is referenced as the matrix attempts to hold sadness and music in the balanced point and counterpoint of spontaneous co-produced "verse". Each line below is spoken by a different person:

- *The power of music.*
- *Music for all occasions.*
- *Does listening to sad music make you depressed?*
- *Isn't beautiful music always sad?*
- *Sometimes we can be addicted to a certain kind of sadness.*
- *It's good pain.*
- *Enveloped in pain.*

The re-casting work of this matrix also reflects the film's affective qualities. Together, the participants produce something that is recognisably "of the film", yet at the same time, a third thing—a collective cultural enactment, original to this particular matrix, and the shared creation of all of its members.

Through this shared enactment, the participants "activate" the aesthetic container that the film itself offers for their experience, and because they find new "forms for feeling" (Langer 1953) within the container, the melancholic affects of the film are transformed. Its sense of devastation is still acknowledged—indeed, it hangs heavy in the room—and at first the matrix appears to "defend itself" by rehearsing the push/pull of antimonies:

- *Letting go.*
- *Holding on.*
- *Discovery.*
- *Reaching out.*
- *Being trapped.*
- *Being sucked in.*
- *Creative freedom.*

However, these fluctuating binaries do not settle into inflexible polarisations. Instead, there is a process of working through with empathy; pain is acknowledged and owned but at stake once again is an ability to occupy a "third area of experiencing" (Winnicott [1971] 1991), neither despair nor "blind hope" but:

- *feeling things intensely with joy and pain at the same time.*

This capacity for ambivalence is hard won and grows throughout the matrix, which registers a seductive drowning "pull" and the excitement that precedes the "release" of the suicidal act:

- *being pulled out in a rip.*

Attuning to this idea—so difficult for those left behind—the matrix seeks an image of at first desperate then benign protection:

- *I keep thinking about how a cattle dog rounds up sheep and when they get desperate because they can't bring everyone together.*
- *And images of toddlers that have those little leads on them so they don't get in trouble, running out and trying to reach the world and then being pulled back for safety from their parents.*

There is a widely held view that cultural material that takes mental distress as its subject matter should end on an optimistic note to prevent "triggering". However well-intentioned, this idea could be misplaced giving rise to a spurious "manic reparation" (Klein 1940). If this were to happen, the matrix, rather than grounding itself in the relational entanglements and ambiguities that the film depicts, would be resorting to "magical

thinking" which seeks to make the world anew and so abolish its pain. When such "solutions" fail, disillusion follows fast and turns to contempt for what had once been idealised—the fall is experienced as a catastrophic collapse of hope, in a world left barren.

Far more useful to this matrix than "manufactured" hopefulness is an acknowledgement of the *conflicting* emotions involved in the loss of a family member through suicide and the growing ease that finally comes with the acceptance of loss:

- *The utter panic of loss.*
- *The helplessness of a parent.*
- *Being lost without a map or a compass.*
- *Riding for miles and miles and then suddenly stopping, then an uncomfortable silence and then growing to be comfortable in that silence.*

In the end, there is a confounding twist in the film's story but no narrative resolution. The temptations of diagnosis, explanation, and message are foregone. The matrix has struggled through its sparse utterances and increasingly generative silences to achieve a sense of connection before the exit:

- *Feeling of recognition at that urge to throw yourself off the edge and the inscrutability of the pianist's face*
- *And then the music helps you escape. . ..*

## 5. Discussion

We have provided examples of how we worked with a psychosocial methodology that had hitherto been used for data collection in research contexts to facilitate audience engagement with two troubling artworks. We suggested that in the particular conditions of shared association provided by the transitional space of a visual matrix, group participants were spontaneously drawn into a process of enactment that in the course of the matrix morphs into play, not merely marking or mirroring but imaginatively taking up and transforming the aesthetic idiom of the work in order to elaborate new imagery and affect. Benjamin (2018) writes of the move from enactment to play (in a clinical setting) as a route whereby a form of paradoxical "thirdness" is achieved—it is the "thirdness that enables the paradox to be accepted and tolerated and respected, and for it not to be resolved. By flight to split off intellectual functioning it is possible to resolve the paradox, but the price of this is the loss of the value of the paradox in itself" (Winnicott [1971] 1991, p. xii).

In a visual matrix, a *partially* dissociated and pre-symbolic "unthought known" is performed (Bollas 1987). In the examples above, it is evident in the rhythmicity and spontaneous temporal flow of contributions to the matrix. Through this performance, or enactment, and in the conditions of containment that the matrix provides, it becomes an object of play. Hence, the enactment is revelatory—its purpose is communication rather than acting out. Its effect is to dissolve the dissociation so that the unthought known presents within an arena of communicative potential. Freed from attribution to a particular speaker, now the property of all the participants in reverie, it sparks a myriad of sensory and visual associations expressed in verbal, metaphorical form. In this process, there is a "rhythmic third" at play, a form of attunement registered in the synchrony of the matrix and in moments when the activity of the matrix appears to be aesthetically isomorphic with the original stimulus (the film). These are moments of "surrender to the third" (Benjamin 2018) in which new symbolisations can occur and find expression in shared affect and imagery. Besides rhythmicity, we see the isomorphism present as palette—the colours of the English seaside in *dis/ordered*, or in *The Invisible Edge*, the red rot of august leather-bound volumes, or the parched landscape of the Australian countryside. It is also present in the retro "genre" of the 1960s (swaying acid-house and bar heaters or the post-colonial architecture of the rural farmstead). Rhythm and tonality infuse acts of re-symbolisation so that the metabolising of the film material leads to the collaborative weaving of new imagistic webs of association, which nevertheless retain aesthetic links to the film.

There is a reparative force in this shared creativity replete with sensory/visual metaphor and the pleasure in its communication which appears to deepen appreciation of the works and render them available for use. According to Benjamin (2018) (following McGilchrist), "The transformation of inchoate feeling into metaphor is part of the process of transforming enactment into play in so far as metaphors do not simply reflect thought, are not pre-digested material". Rather, they are "cognitively active", generating "truly new" links between formally disconnected material in the implicit domain (McGilchrist 2009, p. 179). However, the matrix is not simply a seamless flow. There are minor moments of discontinuity and disruption which demand effort on the part of the participants and are constantly repaired through the paralinguistic meta-communications between them. This helps to keep the matrix working together within the third and through this process, there is an iterative re-vitalisation of symbol with affect.

## 6. Conclusions

Participants in these visual matrices attest to their satisfaction not only with the depth of engagement with the artwork that it facilitates, but in the sense of communion experienced and the new thoughts that arise. In Winnicottian terms, the visual matrix creates the conditions of potential or "third" space in which transitional phenomena proliferate—precursors to symbolisation. We have clues to this happening in the readiness with which participants bring personal biographical material into relation with a shared cultural production. No personal questions are asked in a visual matrix and the focus is on the associative process. Nevertheless, we know that some participants in *The Invisible Edge* had experience of family suicide, because they told us so. We know that the extremely long silences we witnessed involved struggle with the material before people felt able to formulate an idea or present another image. The important thing, however, is that the silences were not merely an indicator of difficulty—indeed, they became progressively more enabling—but that they reflected back the silences in the film, affording space to inhabit its unsettling universe and then to select and re-organise its imagery—even imposing poetic form—so re-creating an aesthetic container for its deathly narrative. This was a case not merely of object relating through identification with the protagonist brothers, who between them held the despair of a family suicide, but of object use (Winnicott [1971] 1991), where the unspeakable experience of suicidality was symbolised, re-elaborated, and released to continue its troubled life in the cultural context of the film, so that the audience could go on with theirs.

In *dis/ordered*, the matrix assigns itself a task not so much of witnessing and reflecting, but of establishing a complex and transcendent solidarity with the narrator by moving beyond the "doer/done to" predicament he portrays. The play in the matrix was fast and flamboyant, producing from the kinaesthetic idiom of the performance a sympathetic rhythmic third (Benjamin 2018). The "borderline" quality and rhythm of childish compulsions "step on a crack and you'll break your mother's back" is owned as a source of enjoyment by the participants, who then riff—with a degree of self-irony—on their love of numbers. In the process, the abusive power of numbers in psychometric testing is subverted—by association with a child's game—while the "compulsive counting" that consigns sufferers to psychiatric classification is ridiculed. Parkinson's self-professed OCD tendencies are transmuted from privatised clinical diagnosis to the pleasurable and proto-artistic compulsions of meaningful social ritual—a symbolic third in Benjamin's (2018) terms. There is also a moral third at play in the participants' efforts—through enactment and transformation to inhabit Parkinson's unsettling world affectively and empathically before taking up his invitation to political critique.

It may be argued that Parkinson's performance itself initiates and enacts this recognition and subversion. To view such work as an individual audience member is touching and inspiring to the degree that it invites connection at an experiential level, if not participation. The visual matrix data, however, points to two important factors in the participants' process: Firstly, it highlights the ease with which an audience enacts connection *in an aesthetic mode*,

influenced by the form, rhythm, and flow as well as content of the artwork (an insight that is likely to be generalisable to other art-viewing experiences). Understanding the nature of such pre-reflective and multimodal engagement is the key to understanding the effectiveness of art in transmitting, holding, and processing affect. Secondly, it points to the potential for collaborative or community use of the creative arts in a reparative or quasi-therapeutic conversation—an engagement that creates the conditions for new experience (Reis 2019). Participants' comments in the post-matrix discussion indicate that there is a perceived value in such collaborative experience, and that images that are upsetting or "triggering" may indeed be creatively processed in group work:

– *I think emotionally if I'd just watched that on my own, I might have gone away and been really upset and felt alone but because we talked about it in a group and I realised everyone had the same—like everyone's dealing with the same shit. So, I think that's where the warm glow comes from, we're all in it together, we're not alone.*

– *That's right, we talked about the emotions. Although we feel desolate but there is still a warm glow, that is the first thing that comes out.*

We consider that the movement from enactment to play in our visual matrices are reparative in relation to the troubling subject matter portrayed because of the transformative power of an embodied response in a shared setting which is intrinsic to the re-symbolisation process. This is not a trivial point if troubling artworks are to be used in mental health contexts. Although the participants speak their associations, they also perform them in co-created presentational symbolisations (Langer 1953) that do not resolve into words. While the primary material of the film and performance takes trauma as its subject matter, reparation arises from shared aesthetic pleasures, deep engagement, and generativity.

**Author Contributions:** Data collection, analysis and authorship were jointly conducted. All authors have read and agreed to the published version of the manuscript.

**Funding:** This research was funded by the Australian Research Council's Linkage Grant Scheme; grant number LP150100481.

**Institutional Review Board Statement:** All subjects received a detailed information sheet and gave written consent prior to participation in this study, which was approved by the Ethics Committee of the University of New South Wales: UNSW HC15513.

**Informed Consent Statement:** Informed consent was obtained from all subjects involved in the study.

**Data Availability Statement:** Data is stored by the University of New South Wales in the form of audio recordings and transcripts. The latter are available on request.

**Conflicts of Interest:** The authors declare no conflict of interest arising in the course of the study or this publication.

## Note

1. Social Dreaming was developed by Gordon Lawrence (2005). Social dreaming uses a matrix but until the visual matrix (Froggett et al. 2015) the process had not adapted to meet the standards of empirical visual research with clear interpretive protocols.

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
