# Peer review of "Aesthetic Enactment: Engagement with Art Evoking Traumatic Loss"

_socsci, doi:10.3390/socsci12080437_

Round 1

Reviewer 1 Report

This report is an original contribution to understanding the unconscious dimensions and therapeutic yield of esthetic experiences. The focus on the co-creativity of the "consumer" of art (viewer, reader, listener) is a valuable contribution to psychoanalytic theory, and the matrix technique comes across as an original approach to documenting individual as well as collective mental processing of trauma and grief.

Although the article's language is dense and demanding, the theorizing is largely adequate and rewarding. The references to Winnicott and Benjamin, including the authors' supplements or qualifications based on their observations, are particularly interesting.

The acknowledgment of experiencing pleasure and pain as concurrent affects is of course relevant in the context of this study. However, the formulation about the alleged "excitement that precedes the excitement of the suicidal act" (p. 10, rows 504, 505) is in my judgment an unfounded romantic rendering of what suicidal states generally entail - deletion recommended. 

Similarly there are a few instances of political interpretations of individual working-through processes observed in the material, which leave this reader burdened with the challenge how to understand them. These formulations, too, lower the worth of this otherwise highly readable and valuable manuscript.  

Author Response

Responses in Italics

This report is an original contribution to understanding the unconscious dimensions and therapeutic yield of esthetic experiences. The focus on the co-creativity of the "consumer" of art (viewer, reader, listener) is a valuable contribution to psychoanalytic theory, and the matrix technique comes across as an original approach to documenting individual as well as collective mental processing of trauma and grief.

Although the article's language is dense and demanding, the theorizing is largely adequate and rewarding.

We have made minor sub-edits throughout to simplify language where this is possible

The references to Winnicott and Benjamin, including the authors' supplements or qualifications based on their observations, are particularly interesting.

The acknowledgment of experiencing pleasure and pain as concurrent affects is of course relevant in the context of this study. However, the formulation about the alleged "excitement that precedes the excitement of the suicidal act" (p. 10, rows 504, 505) is in my judgment an unfounded romantic rendering of what suicidal states generally entail - deletion recommended. 

This has been deleted

Similarly there are a few instances of political interpretations of individual working-through processes observed in the material, which leave this reader burdened with the challenge how to understand them. These formulations, too, lower the worth of this otherwise highly readable and valuable manuscript

We assuming this refers mainly to the first section and have made a minor amendment. However, the political content was part of Clive Parkinson’s presentation and we do point out that the audience were less responsive to this dimension and that how the politics were delivered was as important as its overt content.  However, it was part of the performance and did impact on the visual matrix data. The material cannot be modified or expunged

Reviewer 2 Report

The current research article seeks to employ the visual matrix method as a means of transforming participants' experiences. The concept is intriguing and novel. To investigate concepts such as suicide and trauma, the researcher utilises the festival setting in a novel way.

Following are some remarks regarding the article. The accompanying comments may be used to enhance the manuscript.

1.     The abstract does not adequately describe the research's context, objective, and methodology. It can be rewritten to convey the concept.

2.     The article's title mentions the extremely significant concepts of suicide and trauma, but the introduction section makes no mention of them. Both trauma and suicide are distinct concepts. A brief section in the introduction could explain why these two constructs were conceived.

3.     The concept of loss is distinct from that of suicide. The subjective nature of loss can be distressing to the individual. Consequently, trauma, mental distress, and loss are more suitable concepts than suicide.

4.     The views conveyed in lines 45 to 47 are inherently contradictory. Reparation (transformation) are outcome of specific phases which participants may go through and which may involve many steps.

5.     Line 50 mention about sensory experiences. Author may specify whether physical sensation and emotional response are taken into account and how.

6.     Unconscious enaction- on line 63 may be susceptible to bias. The author can describe how it was measured throughout the entire procedure.

7.     Line 96 lacks clarity.

8.     The phrase traumatic experience is used in line 117. It would be preferable to use consistent terminology throughout.

9.     In line 346, the word "depressed" could be used as a label. As such, when writing scientific papers, it is essential not to make assumptions. Depression is a diagnosis that can be provided by a medical professional. Some specific words for emotions such as "sad" and "gloomy" can be used here.

10.  The manuscript may include a scientific context of the mental health field in order to effectively relate the work to the specific concept.

11.  Numerous art therapies are utilised and well-established for individuals with mental health problems. How the current research can be placed along with such therapy can be a useful addition for the reader.

12.  The art utilised in this instance was very specific and intense. In this context, it is crucial that the researcher be trained to cope with any type of emotional distress. A specific notation on the same will add authenticity of the work.

13.  In visual research matrix analysis, one of the best additions is the researcher's discussion. A similar note can be appended once more.

14.  The term reparation appears throughout the entire manuscript. In evidence-based psychotherapies, such changes are long-lasting and require both the therapist's expertise and the participants' willingness. Due to the fact that such topics have not yet been investigated in the current matrix, it is crucial not to place excessive expectations on the visual matrix.

15.  The visual matrix necessitates participant engagement and introspection; consequently, it is essential to offer support to participants even after the session has concluded. This section may include a description of how the participants were supported, including whether they were provided with support service contacts or facilities.

16.  Adding flowcharts or a visual presentation of results (either pre or post matrix) can make this work more comprehensible for the reader as originally presented by the author.

Author Response

Responses in Italics

  1. The abstract does not adequately describe the research's context, objective, and methodology. It can be rewritten to convey the concept.

The abstract has been re-written along these lines

  1. The article's title mentions the extremely significant concepts of suicide and trauma, but the introduction section makes no mention of them. Both trauma and suicide are distinct concepts. A brief section in the introduction could explain why these two constructs were conceived.

The primary focus is now on traumatic loss and the title has been amended

  1. The concept of loss is distinct from that of suicide. The subjective nature of loss can be distressing to the individual. Consequently, trauma, mental distress, and loss are more suitable concepts than suicide.

Revised accordingly. Clearly suicide is mentioned as it was the subject matter of the film but the primary focus is on traumatic loss

  1. The views conveyed in lines 45 to 47 are inherently contradictory. Reparation (transformation) are outcome of specific phases which participants may go through and which may involve many steps.

We are referring here to ‘reparative potentials' rather than a completed reparation process but we have removed the words “so as to understand better the reparative possibilities of aesthetic engagement” and the passage now refers only to capturing/registering affect. Elsewhere we have now made it clear that we are referring only to the short-term reparative feelings evoked by the experience and make no claims about longer term reparation. However our stance is consistent with Hannah Segal’s (1991) work on art as reparation and this is now referenced (line 40)

  1. Line 50 mention about sensory experiences. Author may specify whether physical sensation and emotional response are taken into account and how.

We have added the following:

As facilitators we are also participants in the matrix and observe the para-linguistic affective-sensory responses and afterwards we listen to the recordings attentively and annotate the transcript of the matrix for purposes of panel interpretation. Emotional states are registered as gesture, tone and facial expression and well as the through the performative dimension of utterances and the silences between them. Examples of this will follow as we attend to each of the matrices in turn. 

  1. Unconscious enaction- on line 63 may be susceptible to bias. The author can describe how it was measured throughout the entire procedure.

Clearly this is a qualitative study within an interpretive paradigm. We have added a new section (2.2) on analysis in an attempt to show how researcher subjectivity was actively deployed while interpretations are moderated. This is a complex procedural and theoretical discussion which we have now  attempted to include as economically as possible while sign-posting the reader to more elaborate peer reviewed discussions of the issue.

  1. Line 96 lacks clarity.

We have deleted “and we ask whether it is this that produces a particularly reparative experience.” We have attempted to clarify the claims we are making with regard to (short-term) reparative potential elsewhere

  1. The phrase traumatic experience is used in line 117. It would be preferable to use consistent terminology throughout.

Noted, and in this instance traumatic has been deleted as experience will suffice. However, trauma is now the primary concept and the draft has  now been checked for consistency.

  1. In line 346, the word "depressed" could be used as a label. As such, when writing scientific papers, it is essential not to make assumptions. Depression is a diagnosis that can be provided by a medical professional. Some specific words for emotions such as "sad" and "gloomy" can be used here.

Depressed has been removed

  1. The manuscript may include a scientific context of the mental health field in order to effectively relate the work to the specific concept.

We situate this paper in the cultural field and that of public mental health, rather than in the field of clinical intervention  (lines 33-35) and have included a reference (Bennett 2022) which discusses this distinction and rationale in detail.

  1. Numerous art therapies are utilised and well-established for individuals with mental health problems. How the current research can be placed along with such therapy can be a useful addition for the reader.

As above, we make it clear that the Big Anxiety Festival occurs within a cultural field and that of public mental health rather than arts and health conceived as a therapeutic or clinical intervention.

Reference included

Bennett, J.; Kenning, G.; Gitau, L.; Moran, R.; Wobcke, M. (2022)Transforming Trauma through an Arts Festival: A Psychosocial Case Study. Soc. Sci. 202312, 249. https://doi.org/10.3390/socsci12040249).

  1. The art utilised in this instance was very specific and intense. In this context, it is crucial that the researcher be trained to cope with any type of emotional distress. A specific notation on the same will add authenticity of the work.

We have added add that the lead facilitator was trained as a mental health social worker. Also that the film on suicide was screened at the  Black Dog Institute which specialises in therapeutic support services for patients is a range of mental heath problems.  The festival has its own protocols in place to ensure safety and support.

See intro/lines 61-66—mention of protocol and support setting

  1. In visual research matrix analysis, one of the best additions is the researcher's discussion. A similar note can be appended once more.

In visual matrix analysis the interpretive frame is established by the participants themselves in the post-matrix discussion of which the researchers are a part. Much of the discussion material is woven into the presentation of the matrices which are considered in substantive, performative and explanatory dimensions as they are in the analysis. For this reason the ‘results’ and ‘discussion’ are not separated.  In an attempt to clarify this we have added section 2.2 and strengthened our remarks on reflexivity.

  1. The term reparation appears throughout the entire manuscript. In evidence-based psychotherapies, such changes are long-lasting and require both the therapist's expertise and the participants' willingness. Due to the fact that such topics have not yet been investigated in the current matrix, it is crucial not to place excessive expectations on the visual matrix.

We have now made it clear throughout that any reparative effects are immediate and we can make no claims for longer term reparation. However, we highlight the role of the matrix in facilitating symbolisation, which within the theoretical frame of Klein and Winnicott that we adopt has the function of finding form that is adequate to feeling – hence making it an object of thought that can be sustained and shared. We regard this sharing, facilitated by the matrix itself, as the beginning of a reparative process which produces further symbolisation in a shared creative production. This is consistent with Hannah Segal’s view of art as reparation (1991), now referenced.

  1. The visual matrix necessitates participant engagement and introspection; consequently, it is essential to offer support to participants even after the session has concluded. This section may include a description of how the participants were supported, including whether they were provided with support service contacts or facilities.

As above (see response to point 12.)

  1. Adding flowcharts or a visual presentation of results (either pre or post matrix) can make this work more comprehensible for the reader as originally presented by the author.

We have carefully considered this suggestion and we think that a flow chart would misrepresent the visual matrix process which is subjected to a depth hermeneutic interpretation. We have a graphic representation of this in the form of a hermeneutic spiral, but to introduce it here would be to burden the text with a methodological discussion which can be found elsewhere and was not the primary aim of this paper. We have opted instead to signpost sources where this discussion on how to extract findings from the matrix can be found (including in graphic form). We hope this is acceptable.